# Predisposing Factors for Severe Complications after Cataract Surgery: A Nationwide Population-Based Study

**DOI:** 10.3390/jcm10153336

**Published:** 2021-07-28

**Authors:** I-Hung Lin, Chia-Yi Lee, Jiann-Torng Chen, Yi-Hao Chen, Chi-Hsiang Chung, Chien-An Sun, Wu-Chien Chien, Hung-Chi Chen, Ching-Long Chen

**Affiliations:** 1Department of Ophthalmology, Tri-Service General Hospital, National Defense Medical Center, Taipei City 11490, Taiwan; petercard@gmail.com (I.-H.L.); jt66chen@gmail.com (J.-T.C.); doc30879@mail.ndmctsgh.edu.tw (Y.-H.C.); 2Department of Ophthalmology, Show Chwan Memorial Hospital, Changhua 50093, Taiwan; ao6u.3msn@hotmail.com; 3Department of Medical Research, Tri-Service General Hospital, National Defense Medical Center, Taipei City 11490, Taiwan; g694810042@gmail.com; 4School of Public Health, National Defense Medical Center, Taipei City 11490, Taiwan; 5Taiwanese Injury Prevention and Safety Promotion Association, Taipei City 11490, Taiwan; 6Department of Public Health, College of Medicine, Fu-Jen Catholic University, New Taipei City 24205, Taiwan; 040866@mail.fju.edu.tw; 7Big Data Research Center, College of Medicine, Fu-Jen Catholic University, New Taipei City 24205, Taiwan; 8Graduate Institute of Life Sciences, National Defense Medical Center, Taipei City 11490, Taiwan; 9Department of Ophthalmology, Chang Gung Memorial Hospital, Linkou, Taoyuan 33305, Taiwan; 10Department of Medicine, Chang Gung University College of Medicine, Taoyuan 33302, Taiwan; 11Center for Tissue Engineering, Chang Gung Memorial Hospital, Linkou, Taoyuan 33305, Taiwan

**Keywords:** cataract surgery, nationwide population-based study, postoperative complication, systemic disease

## Abstract

We conducted a retrospective group study to evaluate the potential systemic risk factors for major postoperative complications of cataract surgery. Individuals diagnosed with (*n* = 2046) and without (*n* = 8184) serious complications after cataract surgery were matched 1:4 for age, sex, and index date obtained using Taiwan’s National Health Insurance Research Database. The outcome was defined as at least one new inpatient or outpatient diagnosis of systemic disease one year before the index date. The effect of demographic data on postoperative complications was also analyzed in the multivariable model. Data were analyzed using univariate and multivariate conditional logistic regression models to calculate odds ratios (ORs) and 95% confidence intervals of the risk of developing serious complications. After the entire study interval, the major postoperative complications of cataract surgery were associated with the following systemic diseases: hypertension (adjusted OR (aOR) = 2.329, *p* < 0.001), diabetes mellitus (aOR = 2.818, *p* < 0.001), hyperlipidemia (aOR = 1.702, *p* < 0.001), congestive heart failure (aOR = 2.891, *p* < 0.001), rheumatic disease (aOR = 1.965, *p* < 0.001), and kidney disease needing hemodialysis (aOR = 2.942, *p* < 0.001). Additionally, demographic data including old age, higher urbanization level, higher level of care, and more frequent inpatient department visits were associated with a higher rate of postoperative complications. In conclusion, metabolic syndrome, chronic heart failure, end-stage renal disease, rheumatic disease, older age, and frequent inpatient department visits are correlated with the development of severe postoperative complications of cataract surgery. Therefore, cataract surgery patients should be informed about a higher possibility of postoperative complications.

## 1. Introduction

Cataract is the leading cause of treatable blindness worldwide [1]. The prevalence of clinically significant cataract is about 93 percent in patients aged 80 or older and influenced the visual function of approximately 95 million people in 2014 [2]. The current management of visually disturbed cataract is via surgery, such as phacoemulsification or manual small-incision cataract surgery [3,4], and the femtosecond laser device can be applied in certain cases [5,6]. Although the general success rate is high for cataract surgery, certain visual-threatening complications can still occur, including cystoid macular edema, pseudophakic bullous keratopathy, retinal detachment, and postoperative endophthalmitis [1,2,7,8].

Several ophthalmic conditions can contribute to a significantly higher rate of postoperative complications in cataract surgery. Intraoperative complications are a common risk factor for postoperative complications. Posterior capsular rupture with or without vitreous loss during cataract surgery is associated with a higher rate of retinal detachment, cystoid macular edema, and postoperative endophthalmitis [8,9,10]. Moreover, other negatively intraoperative statuses, including thermal damage and prolonged phacoemulsification time, can also lead to the development of pseudophakic bullous keratopathy [7]. Prolonged phacoemulsification time is associated with a higher rate of intraoperative complications such as posterior capsular rupture or zonular desinsertion [11]. In addition to intraoperative complications, pre-existing ocular disorders such as uveitis, glaucoma, refractive errors, and a small pupil status are also related to postoperative complications [9,12,13,14].

In addition to ocular conditions, certain systemic disorders are related to the postoperative complications of cataract surgery [10,15]. Males display a higher rate of retinal detachment and postoperative endophthalmitis after cataract surgery [9,16,17]. Diabetes mellitus (DM) is another well-established risk factor for postoperative complications and is associated with cystoid macular edema and postoperative endophthalmitis [12,16,18]. However, few studies have discussed whether other systemic diseases relate to the universal development of postoperative complications. One study revealed the incidence of postoperative endophthalmitis in a patient with hypertension and chronic kidney disease [10], and another study revealed the incidence of postoperative suprachoroidal hemorrhage in a patient with cardiovascular disease, peripheral vascular disease, and hyperlipidemia [19]. Since the presence of metabolic syndrome, chronic kidney disease, and autoimmune disorders can impair ocular condition [20,21,22], it is possible that related systemic diseases serve as a risk factor for postoperative complications of cataract surgery in which the pre-existing systemic disorders cause the eye to be more vulnerable to injury and associate with a higher rate of postoperative complications due to the stress added by cataract surgery. However, this hypothesis requires further elucidation.

Herein, we aimed to evaluate the potential systemic risk factors for major postoperative complications of cataract surgery via the National Health Insurance Research Database of Taiwan. Additionally, the effect of demographic data on postoperative complications was analyzed in a multivariable model.

## 2. Materials and Methods

### 2.1. Data Sources

The National Health Insurance (NHI) program was launched in Taiwan in 1995. As of June 2009, it included contracts with 97% of medical providers with approximately 23 million beneficiaries, including more than 99% of the entire population. The NHI Research Database (NHIRD) uses the *International Classification of Diseases, 9th Revision, Clinical Modification* (ICD-9-CM) codes to record diagnoses. Several studies have demonstrated the accuracy and validity of the diagnoses in the NHIRD [23,24,25]. This study was conducted using the Longitudinal Health Insurance Database (LHID), which is a two-million randomized dataset retrieved from the NHIRD. The LHID contains all medical claims from the outpatient, inpatient, and emergency departments. To protect privacy and ensure data security, the National Health Research Institute encrypts personal identifiers in the LHID before releasing the database.

### 2.2. Study Population

The study population included a group of patients with serious complications after cataract surgery (case group) and another without them (control group). All individuals diagnosed with serious complication after cataract surgery between 2000 and 2015 in the LHID were included in the case group. The diagnosis of serious complication after cataract surgery was confirmed if there was at least one outpatient diagnosis of the following condition by an ophthalmologist during the year following cataract surgery: (1) enucleation of the eye, (2) infectious endophthalmitis, (3) infectious keratitis, (4) bullous keratopathy, (5) perforated corneal ulcer, (6) hyphema, (7) glaucoma, (8) choroidal hemorrhage, (9) cystoid macular edema, (10) retinal break or detachment, (11) Descemet’s membrane detachments, and (12) anterior uveitis. We defined the date of receiving cataract surgery as the index date. Patients with the following conditions during the study period were excluded: (1) blindness, (2) malignant neoplasm of eye, (3) enucleation of the other eye before cataract surgery, (4) serious ophthalmic injury before cataract surgery, (5) receiving another cataract surgery before serious complications, (6) glaucoma before cataract surgery, (7) uveitis before cataract surgery, (8) age < 20 years, and (9) sex unknown. The control group was also selected from the LHID and matched 1:4 with the case group in terms of age, sex, and index date. Patients also received cataract surgery on the index date but had no serious complication within one year after the index date. The same exclusion criteria were used in the control group. Finally, 3915 individuals met the inclusion criteria for the case group. Among them, 1869 individuals were excluded based on exclusion criteria. Therefore, 2046 patients were enrolled as the study group, and another 8184 individuals were selected from the LHID as the control group (Figure 1).

### 2.3. Outcome Measures

The outcome was defined as at least one new inpatient or outpatient diagnosis of the following systemic diseases one year before the index date: (1) hypertension (HTN), (2) DM, (3) ischemic heart diseases (IHD), (4) hyperlipidemia, (5) congestive heart failure (CHF), (6) peripheral vascular disease (PVD), (7) cerebrovascular disease (CVD), (8) allergic pulmonary disease, (9) rheumatic disease (RD), (10) end-stage renal disease (ESRD) that need hemodialysis, (11) purpura and related hemorrhagic diseases, (12) allergic otolaryngologic disease, and (13) allergic dermatological disease. We compared the difference of outcomes between the study and control groups. For the demographic data, we retrieved information on baseline characteristics and clinical details (Table 1) according to the ICD-9-CM, along with the procedure and prescription codes from outpatient and inpatient reimbursement claims in the LHID. Information on income and urbanization level of the patient’s area of residence was considered indicative of socioeconomic status. Income was categorized into three levels (New Taiwan dollars ≥35,000, 18,000–34,999, and <18,000) based on income-related NHI premiums. To eliminate the possible confounding effect of healthcare use, we calculated the average number of outpatient department (OPD) visits, emergency department (ED) visits, and inpatient department (IPD) visits (hospitalizations) per year for each subject on the index date.

### 2.4. Statistical Analyses

Descriptive analysis with the mean and standard deviation (SD) was used to present the differences between the two groups. Continuous variables were compared using *t*-tests, and categorical variables using chi-square tests. Univariate and multivariate conditional logistic regression models were used to calculate odds ratios (ORs) and 95% confidence intervals of the risk of developing serious complication. When performing multivariate conditional logistic regression analyses, all demographic data listed in Table 1 were adjusted to avoid possible confounding effects. For continuous variables, the values of the variables were directly included for adjustment in the regression models; for categorical variables, each variable was treated as a separate dummy variable in the regression models. A two-sided *p* value < 0.05 was considered statistically significant. We performed the analyses using the IBM Statistical Product and Service Solutions for Windows, Version 22.0 (IBM Corp., Armonk, NY, USA).

## 3. Results

### 3.1. Subject Characteristics

A total of 10,230 subjects were enrolled in this study: 2046 in the case group and 8184 in the control group. The mean age of the case group was 46.60 ± 18.84 (SD) years, and the mean age of the control group was 46.52 ± 18.71 years. The proportion of males was 54.9% and the proportion of females was 45.1% in both case and control groups. The subjects were exactly matched for age and sex. Regarding other baseline characteristics, the study group displayed a higher level of medical care and a higher urbanization level than that of the control group. The average number of outpatient visits, ED visits, and hospitalizations per year was also higher. There were no significant differences between the two groups regarding income level (Table 1). In terms of serious complications in the case group, glaucoma was the most frequent (*n* = 803, 39.25%), followed by infectious keratitis (*n* = 311, 15.20%), infectious endophthalmitis (*n* = 256, 12.51%), and bullous keratopathy (*n* = 241, 11.78%) (Table 2).

### 3.2. Comparisons of Systemic Disease with and without Serious Complications after Cataract Surgery

The number of patients with or without systemic disease one year before cataract surgery in the case and control groups is described in Table 3. Diagnosis of serious complication after cataract surgery was significantly associated with having the following systemic disease one year before cataract surgery, as assessed by multivariable conditional logistic regression models: hypertension (adjusted OR (aOR) = 2.329, *p* < 0.001), DM (aOR = 2.818, *p* < 0.001), hyperlipidemia (aOR = 1.702, *p* < 0.001), CHF (aOR = 2.891, *p* < 0.001), RD (aOR = 1.965, *p* < 0.001), and ESRD needing hemodialysis (aOR = 2.942, *p* < 0.001). However, it was not associated with the following: ischemic heart disease (aOR = 1.374, *p* = 0.229), PVD (aOR = 1.310, *p* = 0.305), cerebrovascular disease (aOR = 1.337, *p* = 0.297), allergic pulmonary disease (aOR = 1.406, *p* = 0.233), purpura and related hemorrhagic diseases (aOR = 1.503, *p* = 0.165), allergic otolaryngologic disease (aOR = 1.342, *p* = 0.284), or allergic dermatological disease (aOR = 1.143, *p* = 0.231). The diagnosis of serious complications after cataract surgery was also significantly associated with having any one of the above potential risk factors (aOR = 2.008, *p* < 0.001) (Table 4). These results can be seen in the forest plot (Figure 2).

### 3.3. Comparisons of Demographic Data with and without Serious Complications after Cataract Surgery

As analyzed through multivariable conditional logistic regression models, the diagnosis of serious complications after cataract surgery was significantly associated with the following demographic data on the index date: higher age (using age 20–29 as reference, age 40–49: aOR = 1.251, *p* < 0.001; age 50–59: aOR = 1.276, *p* < 0.001; age 60–69: aOR = 1.488, *p* < 0.001; age 70–79: aOR = 1.330, *p* < 0.001; age ≥ 80: aOR = 1.392, *p* < 0.001), higher urbanization level (using lowest level 4 as reference, level 3: aOR = 1.835, *p* = 0.036; level 2: aOR = 2.703, *p* < 0.001; level 1: aOR = 2.975, *p* < 0.001), higher level of care (using local hospital as reference, regional hospital: aOR = 2.286, *p* < 0.001; hospital center: aOR = 2.896, *p* < 0.001), and more frequent IPD visit (aOR = 1.277, *p* = 0.003). However, it was not associated with young age, such as 30–39 (aOR = 1.098, *p* = 0.072), more frequent OPD visits (aOR = 1.104, *p* = 0.306), more frequent ED visits (aOR = 1.156, *p* = 0.227), or different income level (using insured premium < 18,000 as reference, insured premium 18,000–34,999: aOR = 0.981, *p* = 0.701; insured premium ≥35,000: aOR = 0.875, *p* = 0.652) (Table 5).

## 4. Discussion

The current study revealed a significant correlation between major postoperative complications of cataract surgery and systemic diseases such as hypertension, DM, hyperlipidemia, CHF, RD, and ESRD receiving hemodialysis. Demographic data, including old age, higher urbanization level, higher level of care, and more frequent IPD visits were associated with a higher rate of postoperative complications. The above are independent risk factors for the occurrence of postoperative complications according to the multivariable analysis.

The DM is a component of metabolic syndrome, and the presence of DM correlates with the development of postoperative complications in cataract surgery [15,18,26]. In a previous study, DM was associated with a higher rate of cystoid macular edema [8], and other studies demonstrated that DM patients were prone to develop postoperative endophthalmitis [10,27]. In the current study, DM is also a significant risk factor for major postoperative complications of cataract surgery with a prominent aOR of 2.818, consistent with the results from previous research. DM is associated with vascular damage, impaired immune status, an elevated inflammation reaction, and poor wound healing status [28,29,30,31], and the postoperative complications of orthopedic surgery were also elevated in DM patients [32]. Thus, the eyes of such a population may be more vulnerable to external damage and more prone to postoperative complications, as supported by the results of the current study. Except for DM, other metabolic diseases, including hypertension and dyslipidemia, and hyperlipidemia, also correlated with the development of postoperative complications in the current study. Hypertension may lead to poor postoperative status, including delay wound healing in root canal treatment and a higher rate of postoperative endophthalmitis [10,33]. Conversely, the dyslipidemia status can damage the vascular endothelium and lead to the formation of atherosclerotic plaque [34]. The weakened ocular vasculature originating from the above dyslipidemia-related morbidities may allow the infection, inflammation, and hemorrhagic complications to occur more frequently. Ling et al. also found that the rate of postoperative suprachoroidal hemorrhage was higher in hyperlipidemia patients [19]. However, IHD did not display a higher rate of postoperative complications in the current study. Ling et al. also found that postoperative suprachoroidal hemorrhage was not associated with IHD [19]. A possible reason is that the types of IHD included in the current study were acute coronary disease and myocardial infarction, acute events that do not cause long-term influences in ocular condition.

Except for metabolic syndrome, the systemic risk factors for postoperative complications in cataract surgery include CHF, ESRD with hemodialysis, and RD, which have been seldom reported elsewhere. Chronic renal disease can lead to fluid retention and decreased immune condition [35,36], which may make certain postoperative complications such as cystoid macular edema, bullous keratopathy, rhegmatogenous retinal detachment, and infectious diseases occur with a higher incidence. In addition, a previous study stated that patients receiving hemodialysis displayed a higher rate of developing subsequent glaucoma [21], which was also enrolled in the postoperative complications in the current study. However, RD presented an elevated autoimmune activity and inflammatory reaction [37], and previous data demonstrated a higher rate of the anterior uveitis in the patient with RD [38]. Moreover, RD and glaucoma shared certain pathogenesis characteristics, including the involvement of CD4 T-cells and microbiota [39], thus the presence of RA may increase the possibility of glaucoma development. CHF can induce fluid retention similar to ESRD [40] and may cause postoperative complications such as ESRD.

Regarding demographic data, older age is a prominent risk factor for postoperative complications. It is reasonable that older patients are more susceptive to postoperative complications after cataract surgery with a more degenerated eyeball, and the rates of postoperative endophthalmitis and endotheliitis are also higher in older populations [27,41]. In Taiwan, secondary and tertiary hospitals received many patients with postoperative complications after cataract surgery since they have the equipment and specialists to handle these severe morbidities. Additionally, secondary and tertiary hospitals are often located in metropolitan areas. The above two reasons may explain the higher rate of postoperative complications in the regional hospital and medical center and in the higher urbanization area in the current study. Conversely, patients visiting the IPD more frequently were associated with a higher rate of postoperative complications. We think a possible explanation for this phenomenon is that patients frequently visiting the IPD may be admitted to the general ward or intensive care unit, and their general condition may be worse compared to those visiting the OPD or ED whose disorders should be mild or transient. Accordingly, patients frequently visiting the IPD may display a higher rate of ocular morbidities after surgical intervention. The male sex was not a significant risk factor for postoperative complications in the current study, contrary to previous experiences [9,10,16,42]. We speculated that postoperative complications included in the current study contain female-predominant diseases such as glaucoma and certain types of uveitis [43,44], thus the ratio of male patients in such diseases was lower. In addition, males are more commonly associated with metabolic disease [45], thus the effect of the male sex may be diminished in the multivariable analysis while considering the effect of metabolic syndrome due to collinearity.

Epidemiologically, the overall incidence of severe postoperative complications after cataract surgery in the current study is about 0.1 percent. This incidence is slightly higher than the ratio reported in previous population-based research in Eastern Asia [10]. However, previous studies only included postoperative endophthalmitis, while the current study enrolled 12 types of postoperative complications [10]. The most common postoperative complications in the current study are glaucoma, followed by infectious keratitis, infectious endophthalmitis, and pseudophakic bullous keratopathy. The percentages of infectious diseases and pseudophakic bullous keratopathy were similar to a previous study [2], while the higher incidence of glaucoma is contrary to previous findings [46]. We speculate two possible reasons for the high rate of postoperative glaucoma in the current study. Firstly, the development of acute intraocular pressure spike, pseudophakic pupillary block, and malignant glaucoma during the postoperative period may account for the majority of postoperative glaucoma episodes which were also reported in previous studies [47,48]. These acute events may be different from the classic glaucoma associated with a chronic or persistent disease course, but they can definitively lead to optic nerve damage. Thus, we enrolled them into the glaucomatous disorders in the current study. It seems that the postoperative glaucoma accounts for the majority of postoperative complications, but the incidence of postoperative glaucoma is approximately 0.8 percent in the entire population receiving cataract surgery according to our database, which is a less common disorder. Nonetheless, we admit that patients regarded as postoperative glaucoma may have undiscovered preoperative ocular hypertension since the ocular hypertension related diagnostic codes are not routinely entered in the healthcare system in Taiwan because seldom treatment is needed for these patients. Since glaucoma can lead to irreversible blindness and account for a large proportion of postoperative complications in our current study [44], aggressive postoperative anti-glaucomatous management should be applied to patients with predisposing factors for severe postoperative complications of cataract surgery.

There are certain limitations to the current study. First, the retrospective design can cause the heterogeneity of study population to be higher than the prospective design. Second, the use of claimed data rather than real medical documents for data collection can cause the severity of disease and the associated laboratory exam values (especially HbA1c and LDL, which can serve as real risk factors and severity markers for DM and hyperlipidemia), and ocular images to be inaccessible. The related surgery information such as operating time, surgeon’s experience, and preoperative refraction were also inaccessible. The intraoperative conditions and intraoperative complications were also unavailable due to either the absence of diagnostic codes or the diagnostic codes for the disorders were too general or indistinguishable. Moreover, we combined all postoperative complications into a single population due to the small numbers in complications, such as hyphema, choroidal hemorrhage, and enucleation, which can produce a significant statistical bias. However, not all the postoperative complications shared similar pathophysiology, and we did not calculate the aOR of each possible risk factor for different postoperative complications. Consequently, the comparison in the current study cannot reflect the exact mechanism of predisposing factors for postoperative complications. Nevertheless, the incidence of any postoperative complications was significantly higher if one of the possible risk factors existed in the current study. Thus, our findings may demonstrate the general risk factors for postoperative complications of cataract surgery.

## 5. Conclusions

In conclusion, the presence of hypertension, DM, hyperlipidemia, CHF, RD, and ESRD correlated with the development of severe postoperative complications of cataract surgery after adjusting for multiple potential confounders. Furthermore, older age and frequent IPD visits shared a similar relationship. Consequently, physicians may recognize patients with predisposing risk factors of severe postoperative complications before cataract surgery, and patients should be informed about their higher possibility of postoperative complications. Further large-scale prospective studies to evaluate the effects of systemic diseases, especially metabolic syndrome, on the development and severity of each severe postoperative complication are granted.

## Figures and Tables

**Figure 1 jcm-10-03336-f001:**
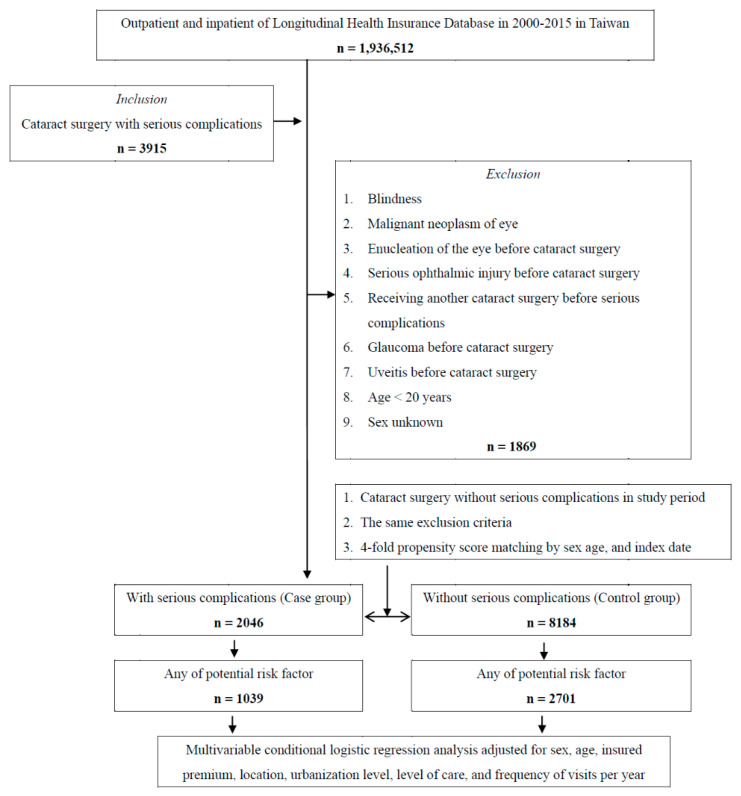
Flowchart of study sample selection.

**Figure 2 jcm-10-03336-f002:**
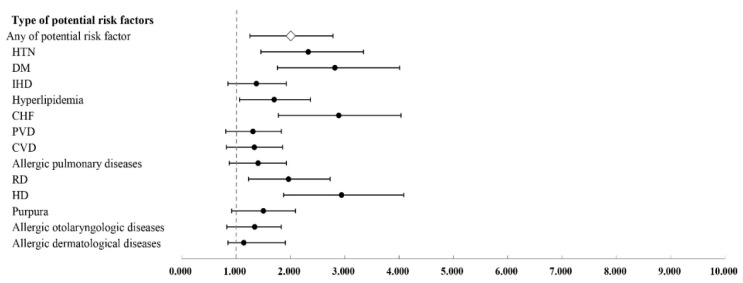
Forest plot for risk factors for cataract surgery with serious complications analyzed using multivariable conditional logistic regression.

**Table 1 jcm-10-03336-t001:** Characteristics of case cohort and control groups.

Variables	Total		Case		Control		*p*
	*n*	%	*n*	%	*n*	%	
**Sex**							0.999
Male	5620	54.94	1124	54.94	4496	54.94	
Female	4610	45.06	922	45.06	3688	45.06	
**Age group (years)**							0.999
20–29	1295	12.66	259	12.66	1036	12.66	
30–39	2415	23.61	483	23.61	1932	23.61	
40–49	2515	24.58	503	24.58	2012	24.58	
50–59	2125	20.77	425	20.77	1700	20.77	
60–69	1170	11.44	234	11.44	936	11.44	
70–79	555	5.43	111	5.43	444	5.43	
≧80	155	1.52	31	1.52	124	1.52	
**Insured premium (NT$)**							0.304
<18,000	6667	65.17	1315	64.27	5352	65.40	
18,000–34,999	2516	24.59	503	24.58	2013	24.60	
≧35,000	1047	10.23	228	11.14	819	10.01	
**Urbanization level**							<0.001 *
1 (Highest)	3915	38.27	903	44.13	3012	36.80	
2	3626	35.44	823	40.22	2803	34.25	
3	1446	14.13	211	10.31	1235	15.09	
4 (Lowest)	1243	12.15	109	5.33	1134	13.86	
**Level of care**							<0.001 *
Hospital center	6145	60.07	1356	66.28	4789	58.52	
Regional hospital	2345	22.92	442	21.60	1903	23.25	
Local hospital	1740	17.01	248	12.12	1492	18.23	
**Frequency of visits per year**							
OPD	4.57 ± 4.26		5.86 ± 4.89		4.25 ± 4.02		<0.001 *
ED	1.12 ± 1.21		1.27 ± 1.34		1.08 ± 1.17		<0.001 *
IPD	2.04 ± 1.87		2.89 ± 2.76		1.83 ± 1.50		<0.001 *

* denotes significant difference between the case and control groups. OPD: outpatient department; ED: emergency department; IPD: inpatient department; *p:* chi-square/Fisher’s exact test on category variables and *t*-test on continue variables.

**Table 2 jcm-10-03336-t002:** Distribution of serious complications.

Complications	*n*	%
Overall	2046	
Enucleation of the eye	7	0.34
Infectious endophthalmitis	256	12.51
Infectious keratitis	311	15.20
Bullous keratopathy	241	11.78
Perforated corneal ulcer	32	1.56
Hyphema	35	1.71
Glaucoma	803	39.25
Choroidal hemorrhage	33	1.61
Cystoid macular edema	152	7.43
Retinal detachment or defects	87	4.25
Rupture in Descemet’s membrane	2	0.10
Anterior uveitis	130	6.35

**Table 3 jcm-10-03336-t003:** The number of patients with or without systemic diseases one year before cataract surgery in the case and control cohorts.

Serious Complications	Total		Case		Control	
Variables	*n*	%	*n*	%	*n*	%
**Total**	10,230		2046	20.00	8184	80.00
**Any of potential risk factor**						
Without	6490	63.44	1007	49.22	5483	67.00
With	3740	36.56	1039	50.78	2701	33.00
**HTN**						
Without	8321	81.34	1457	71.21	6864	83.87
With	1909	18.66	589	28.79	1320	16.13
**DM**						
Without	7988	78.08	1260	61.58	6728	82.21
With	2242	21.92	786	38.42	1456	17.79
**IHD**						
Without	9635	94.18	1922	93.94	7713	94.24
With	595	5.82	124	6.06	471	5.76
**Hyperlipidemia**						
Without	8888	86.88	1716	83.87	7172	87.63
With	1342	13.12	330	16.13	1012	12.37
**CHF**						
Without	10,129	99.01	2010	98.24	8119	99.21
With	101	0.99	36	1.76	65	0.79
**PVD**						
Without	9685	94.67	1936	94.62	774	94.68
With	545	5.33	110	5.38	435	5.32
**CVD**						
Without	9720	95.01	1942	94.92	7778	95.04
With	510	4.99	104	5.08	406	4.9
**Allergic pulmonary disease allergic dermatological disease**						
Without	8803	86.05	1743	85.19	7060	86.27
With	1427	13.95	303	14.81	1124	13.73
**RD**						
Without	10,124	98.96	2017	98.58	8107	99.06
With	106	1.04	29	1.42	77	0.94
**ESRD**						
Without	9125	89.20	1647	80.50	7478	91.37
With	1105	10.80	399	19.50	706	8.63
**Purpura**						
Without	10,172	99.43	2033	99.36	8139	99.45
With	58	0.57	13	0.64	45	0.55
**Allergic otolaryngologic disease**						
Without	9689	94.71	1935	94.57	7754	94.75
With	541	5.29	111	5.43	430	5.25
**Allergic dermatological disease**						
Without	9807	95.87	1958	95.70	7849	95.91
With	42	4.13	88	4.30	335	4.09

CHF: congestive heart failure; CVD: cerebrovascular disease; DM: diabetes mellitus; ESRD: end-stage renal disease; HTN: hypertension; IHD: ischemic heart disease; PVD: peripheral vascular disease; RD: rheumatic disease.

**Table 4 jcm-10-03336-t004:** Risk factors for cataract surgery with serious complications analyzed using multivariable conditional logistic regression among different exposures.

Potential Risk Factors	Adjusted OR	Upper 95% CI	Lower 95% CI	*p*
**Any potential risk factor**	2.008	1.256	2.785	<0.001 *
HTN	2.329	1.459	3.345	<0.001 *
DM	2.818	1.761	4.011	<0.001 *
IHD	1.374	0.852	1.925	0.229
Hyperlipidemia	1.702	1.063	2.372	<0.001 *
CHF	2.891	1.780	4.038	<0.001 *
PVD	1.310	0.811	1.835	0.305
CVD	1.337	0.824	1.856	0.297
Allergic pulmonary diseases	1.406	0.875	1.927	0.233
RD	1.965	1.230	2.733	<0.001 *
ESRD	2.942	1.875	4.088	<0.001 *
Purpura	1.503	0.921	2.096	0.165
Allergic otolaryngologic disease	1.342	0.831	1.834	0.284
Allergic dermatological disease	1.143	0.852	1.907	0.231

Adjusted OR: adjusted odds ratio, adjusted for the variables listed in Table 3; CI: confidence interval; CHF: congestive heart failure; CVD: cerebrovascular disease; DM: diabetes mellitus; ESRD: end-stage renal disease; HTN: hypertension; IHD: ischemic heart disease; PVD: peripheral vascular disease; RD: rheumatic disease. * denotes significant differences between the case and control groups.

**Table 5 jcm-10-03336-t005:** Risk factors for cataract surgery with serious complications analyzed using multivariable conditional logistic regression among different demographic data.

Variables	Adjusted OR	Upper 95% CI	Lower 95% CI	*p*
**Sex**				
Male	1.555	0.894	2.134	0.201
Female	Reference			
**Age group (years)**				
20–29	Reference			
30–39	1.098	0.976	1.135	0.072
40–49	1.251	1.100	1.378	<0.001 *
50–59	1.276	1.124	1.402	<0.001 *
60–69	1.488	1.301	1.547	<0.001 *
70–79	1.330	1.212	1.488	<0.001 *
≥80	1.392	1.246	1.490	<0.001 *
**Insured premium (NT$)**				
<18,000	Reference			
18,000–34,999	0.981	0.582	1.902	0.701
≥35,000	0.875	0.443	1.804	0.652
**Urbanization level**				
1 (highest)	2.975	1.386	4.278	<0.001 *
2	2.703	1.233	3.902	<0.001 *
3	1.835	1.014	2.801	0.036 *
4 (lowest)	Reference			
**Level of care**				
Hospital center	2.896	1.268	4.065	<0.001 *
Regional hospital	2.286	1.124	3.887	<0.001 *
Local hospital	Reference			
**Frequency of visits per year**				
OPD	1.104	0.775	1.384	0.306
ED	1.156	0.894	1.422	0.227
IPD	1.277	1.043	1.356	0.003 *

Adjusted OR: adjusted odds ratio, adjusted for the variables listed in Table 3; CI: confidence interval; ED: emergency department; IPD: inpatient department; OPD: outpatient department. * denotes significant difference between the case and control groups

## Data Availability

The data that support the findings of this study are available from the National Health Insurance Administration of Taiwan, but restrictions apply to the availability of these data, which were used under license for the current study, and are thus not publicly available. Data are available from the authors upon reasonable request and with permission of the National Health Insurance Administration of Taiwan.

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
