# Peer review of "Predisposing Factors for Severe Complications after Cataract Surgery: A Nationwide Population-Based Study"

_jcm, 2021, doi:10.3390/jcm10153336_

Round 1

Reviewer 1 Report

a very nice paper. i think it should be emphasis more, that medical data like, HbA1c that was not included in the study, my help us distinguish DM severity as a real risk factor. it is also true for other diseases.   

Author Response

Point 1: A very nice paper. I think it should be emphasis more, that medical data like, HbA1c that was not included in the study, may help us distinguish DM severity as a real risk factor. It is also true for other diseases.

Response 1: Thanks for the suggestion. The use of claimed data rather than real medical documents for data collection turned the severity of disease, the associated laboratory exam value especially HbA1c and LDL which can served as real risk factors and severity markers for DM and hyperlipidemia, and ocular image inaccessible. We add this viewpoint at the limitation section which is available at line 327-331.

Line 327-331: Second, the use of claimed data rather than real medical documents for data collection turned the severity of disease, the associated laboratory exam value especially HbA1c and LDL which can served as real risk factors and severity markers for DM and hyperlipidemia, and ocular image inaccessible.

Reviewer 2 Report

This is a well-written population-based comprehensive paper evaluating systemic risk factors for major postoperative complications of cataract surgery. One of the advantages is that the authors identified systemic risk factors not described in the literature before.

Line 53: Should the “Femtosecond” be written with the capital letter?

Line 63: Is prolonged phacoemulsification time an intraoperative complication or is it the reason for complications?

Line 111: Enucleation of the eye before cataract surgery was an exclusion criterion. What does it mean? Is it related to the other eye?

Line 130: Is the intracameral cefuroxime axetil standard of care in Taiwan? Why were allergic dermatological disease considered? As a potential risk for blepharitis? Why was allergic otolaryngologic disease considered?

Line 171 Table2: Is the “Cystoid macular degeneration”  or  “Cystoid macular oedema”?

Line 233: According to the definition: “Metabolic syndrome is a cluster of conditions that occur together, increasing your risk of heart disease, stroke and type 2 diabetes.” So the presence of diabetes does not equal metabolic syndrome.

Line 246: What is the association between wound healing in root canal treatment and major complications after cataract surgery?

Line 260-262: Do the authors write about serous or rhegmatogenous retinal detachment? In line 108 “retinal break or detachment” is named as a mojor complication

Line 296: “silently” or “slightly”?

In the Introduction (lines 58-67) the authors nicely summarize the ophthalmic conditions that increase the risk of the cataract surgery.

a) It would be interesting to know the distribution of these conditions in the studied population. Line 88: Does the NHI Program allow the detection of concomitant ophthalmic diseases (e.g. H44.2 – high myopia; H40-preexisting glaucoma etc. according to ICD-9).

b)Furthermore, e.g. Pseudoexfoliation is known to be related to the cardiovascular diseases. Consequently a question arises, if the major complications are due the tophthalmic or systemic disease?

4.Discussion:

According to the line 113 preexisting glaucoma was an exclusion criterion. Could the authors discuss the reasons for a relatively high rate of postoperative glaucoma (undiagnosed before surgery? Ocular hypertension before surgery? etc) What could be the pathomechanism of cataract surgery leading to glaucoma?

2.Materials and Methods

Postoperative glaucoma presenting almost 40% of major complications is an important issue. Could the authors define the criteria for this diagnosis? Was also increased intraocular pressure 24h after surgery included in this group?

What was the posterior capsule rupture rate in two study groups?

Author Response

Response to Reviewer 2 Comments

Point 1: This is a well-written population-based comprehensive paper evaluating systemic risk factors for major postoperative complications of cataract surgery. One of the advantages is that the authors identified systemic risk factors not described in the literature before.

Response 1: Thank you. We make revision according to your suggestion carefully and hope our effort can response your request well.

Point 2: Line 53: Should the “Femtosecond” be written with the capital letter?

Response 2: Thanks for the suggestion. Actually, it does not need to be written with capital letter. Consequently, we change it to lower case which is available at line 53.

Point 3: Line 63: Is prolonged phacoemulsification time an intraoperative complication or is it the reason for complications?

Response 3: Thanks for the suggestion. Prolonged phacoemulsification time is the reason for complications instead of an intraoperative complication. We admitted the phrase “intraoperative complication” is inappropriate to describe prolonged phacoemulsification time. Thus we change “intraoperative complication” to “negatively intraoperative statuses” which can also depict thermal burn. Also, we added a sentence and a new reference [11] to illustrate the prolonged phacoemulsification time is related to some intraoperative complications during cataract surgery. The new phrase is available at line 62-66.

Line 62-66: Moreover, other negatively intraoperative statuses including thermal damage and prolonged phacoemulsification time can also lead to the development of pseudophakic bullous keratopathy [7]. And the prolonged phacoemulsification time is associated with higher rate of intraoperative complications like posterior capsular rupture or zonular desinsertion [11].

New reference:

  1. Pérez-Campagne, E.; Basdekidou, C.; Petropoulos, I.K.; Noachovitch, B.; Moubri, M. Impact of preoperative and intraoperative factors in cataract surgery. Klin Monbl Augenheilkd 2013, 230, 326-328.

Point 4: Line 111: Enucleation of the eye before cataract surgery was an exclusion criterion. What does it mean? Is it related to the other eye?

Response 4: Yes, it is related to the other eye. Because we want to study the normal population, not the patient with serious eye disease before. For the patient had enucleation of the other eye, it may also have the potential of developing serious eye disease in the eye for cataract surgery, which may become the confounding factor in our study. Therefore, we excluded these patients for getting a more accurate result in our study. We also change” Enucleation of the eye before cataract surgery” to “Enucleation of the other eye before cataract surgery” at line 115-116 to avoid misunderstanding.

Point 5: Line 130: Is the intracameral cefuroxime axetil standard of care in Taiwan? Why were allergic dermatological disease considered? As a potential risk for blepharitis? Why was allergic otolaryngologic disease considered?

Response 5: Intracameral cefuroxime axetil is not the standard of care in Taiwan. Allergic dermatological disease and allergic otolaryngologic disease was considered because we think these kinds of allergic disease may increase the systemic immune response and therefore increase the inflammation of the eye after the cataract surgery, and further increase the risk of postoperative complications. Besides, allergic dermatological disease including allergic blepharitis, which may be a potential risk for infectious blepharitis. Infectious blepharitis is the risk factor for acute postoperative endophthalmitis [a]. However, neither allergic dermatological disease nor allergic otolaryngologic disease show significant association with postoperative complications in our study.

Reference:

  1. Speaker, M. G., Milch, F. A., Shah, M. K., Eisner, W., & Kreiswirth, B. N. Role of external bacterial flora in the pathogenesis of acute postoperative endophthalmitis. Ophthalmology 1991, 98(5), 639–650.

Point 6: Line 171 Table2: Is the “Cystoid macular degeneration” or “Cystoid macular oedema”?

Response 6: It’s a mistake. It should be “Cystoid macular edema” instead of “Cystoid macular degeneration”. We had revised it at Line 175, Table 2. Thanks for your kindly suggestion.

Point 7: Line 233: According to the definition: “Metabolic syndrome is a cluster of conditions that occur together, increasing your risk of heart disease, stroke and type 2 diabetes.” So the presence of diabetes does not equal metabolic syndrome.

Response 7: Thanks for the suggestion. Maybe our poor writing skill lead to some misunderstanding. We re-wrote this sentence which is now available at line 237-238.

Line 237-238: The DM is a component of metabolic syndrome and the presence of DM, would correlates with the development of postoperative complications in cataract surgery [15,18,26].

Point 8: Line 246: What is the association between wound healing in root canal treatment and major complications after cataract surgery?

Response 8: Thanks for the suggestion. In this sentence, we tried to express that hypertension will lead to poor postoperative condition (poor wound healing and infection) rather than demonstrate the association between root canal wound healing and post-cataract surgery complication. We re-wrote the sentence to make it more readable which is available at line 250-252.

Line 250-252: Hypertension could lead to poor postoperative status, including inhibit delay wound healing in root canal treatment, and the higher rate of postoperative endophthalmitis was higher in hypertension patients [10,33].

Point 9: Line 260-262: Do the authors write about serous or rhegmatogenous retinal detachment? In line 108 “retinal break or detachment” is named as a major complication

Response 9: Thanks for the suggestion. We write about rhegmatogenous retinal detachment. We modified the sentence at line 266.

Line 266: …which might make certain postoperative complications like cystoid macular edema, bullous keratopathy, rhegmatogenous retinal detachment, and infectious diseases occur with a higher incidence.

Point 10: Line 296: “silently” or “slightly”?

Response 10: Thanks for the suggestion. We change the “silently” to “slightly” at line 300.

Point 11: In the Introduction (lines 58-67) the authors nicely summarize the ophthalmic conditions that increase the risk of the cataract surgery.

  1. a) It would be interesting to know the distribution of these conditions in the studied population. Line 88: Does the NHI Program allow the detection of concomitant ophthalmic diseases (e.g. H44.2 – high myopia; H40-preexisting glaucoma etc. according to ICD-9).

Response 11: Thanks for the suggestion. However, the intraoperative conditions like thermal damage, prolonged phacoemulsification time and small pupil cannot be found in the claimed database due to absent of related diagnostic codes. Although posterior capsular tear and vitreous loss own ICD diagnostic codes, the corresponded codes are general and indistinguishable thus physician in Taiwan rarely applied them. Still, we admitted the lack of intraoperative information is a shortcoming and we added the description in the limitation section at line 331-333. Besides, the NHI program can detect concomitant ophthalmic diseases and we excluded any type of pre-existing glaucoma in the selection process. The code for high myopia is rarely used in clinical practice in Taiwan due to no related reimbursement in our health-care system, thus we did not considered this disease/code to prevent extreme underestimation and subsequent bias.

Line 331-333: The intraoperative conditions and intraoperative complications are also unavailable due to either the absent of diagnostic codes or the diagnostic codes for the disorders are too general/indistinguishable.

Point 12: b)Furthermore, e.g. Pseudoexfoliation is known to be related to the cardiovascular diseases. Consequently a question arises, if the major complications are due to the ophthalmic or systemic disease?

Response 12: Thanks for the suggestion. We think that both the cataract surgery and systemic diseases could lead to the major postoperative complications. The presence of systemic diseases would put the eye under a greater risk of major morbidity, which could be triggered by the performance of cataract surgery. We excluded some pre-existing major complications like uveitis, glaucoma and major ocular trauma in the selection process, and an ophthalmologist would not arrange cataract surgery unless the previous infectious/inflammation diseases resolve in our knowledge. Consequently, the major complications recorded in the current study are more likely due to the synergic effects of cataract surgery and underlying systemic morbidities rather than a long-lasting ocular disorder related to previous systemic diseases. We now described our concept with more details in the introduction at line 80-85.

Line 80-85: Since the presence of metabolic syndrome, chronic kidney disease, and autoimmune disorders could impair ocular condition [20-22], it is possible that related systemic dis-eases served as a risk factor for postoperative complications of cataract surgery in which the pre-existing systemic disorders make the eye more vulnerable to insult and associate with higher rate of postoperative complications due to the stress added by cataract surgery. However, this hypothesis requires further elucidation.

Point 13: 4.Discussion:

According to the line 113 preexisting glaucoma was an exclusion criterion. Could the authors discuss the reasons for a relatively high rate of postoperative glaucoma (undiagnosed before surgery? Ocular hypertension before surgery? etc) What could be the pathomechanism of cataract surgery leading to glaucoma?

Response 13: Thanks for the suggestion. We speculate the high rate of postoperative glaucoma is due to the development of acute intraocular pressure spike, pseudophakic pupillary block and malignant glaucoma during the postoperative period. It may be different from the classic glaucoma with a chronic/persistent course but it can lead to optic nerve damage thus we still enrolled them into the glaucomatous disorder in the current study. Although the postoperative glaucoma account for the majority of postoperative complications, the incidence of postoperative glaucoma is approximately 0.8 percent in the whole population receiving cataract surgery, which is still a relative low value. Still, we admitted that some patients with postoperative glaucoma may have undiscovered preoperative ocular hypertension since the ocular hypertension-related diagnostic codes are not routinely entered by physician since seldom treatment is needed for these patients. We added these descriptions in the discussion section at line 307-321.

Line 307-321: We speculate two possible reasons for the high rate of postoperative glaucoma in the current study. Firstly, the development of acute intraocular pressure spike, pseudo-phakic pupillary block and malignant glaucoma during the postoperative period may account for the majority of postoperative glaucoma episodes which also been reported in previous study [47,48]. These acute events may be different from the classic glauco-ma that with a chronic or persistent disease course, but they can definitively lead to optic nerve damage thus we enrolled them into the glaucomatous disorders in the current study. It seems that the postoperative glaucoma account for the majority of postoperative complications, but the incidence of postoperative glaucoma is approximately 0.8 percent in the whole population receiving cataract surgery according to our database, which is still a less-common disorder. Still, we admitted that some patients regarded as postoperative glaucoma may have undiscovered preoperative ocular hy-pertension since the ocular hypertension-related diagnostic codes are not routinely entered in the health-care system in Taiwan because seldom treatment is needed for these patients.

New reference:

  1. Harbour, J.W.; Rubsamen, P.E.; Palmberg, P. Pars plana vitrectomy in the management of phakic and pseudophakic malignant glaucoma. Arch Ophthalmol 1996, 114, 1073-1078.
  2. Seth, N.G.; Thattaruthody, F.; Jurangal, A.; Pandav, S.S. Late onset pupillary block glaucoma following phacoemulsification with posterior chamber intraocular lens implantation. Eur J Ophthalmol 2020, 30, Np26-np28.

Point 14: 2.Materials and Methods

Postoperative glaucoma presenting almost 40% of major complications is an important issue. Could the authors define the criteria for this diagnosis? Was also increased intraocular pressure 24h after surgery included in this group?

Response 14: The diagnosis of postoperative glaucoma was retrieved from the Taiwan’s National Health Insurance Research Database, and it include all kinds of glaucoma which was diagnosed and inputted to the database by the ophthalmologists all over the Taiwan. As we described at line 327-331, the use of claimed data rather than real medical documents for data collection turned the severity of disease, the associated laboratory exam value, and ocular image inaccessible. Therefore, we cannot get the result of intraocular pressure, gonioscopy, optical coherence tomography and visual field exam which was needed for diagnosis and classification of glaucoma, so we could not directly approach the subjects to confirm their diagnosis. For the patients with increased intraocular pressure 24h after surgery, most ophthalmologists in Taiwan may use “ocular hypertension” as the record of diagnoses instead of “glaucoma”. Therefore, these patients were not included in our study.

Point 15: What was the posterior capsule rupture rate in two study groups?

Response 15: If the patient had the posterior capsule rupture, most of them would receive another cataract surgery for secondary intraocular lens implantation. But as we described at line 116-117, the patients who had received another cataract surgery before serious complications was excluded. Therefore, most patients with posterior capsule rupture were exclude in the two study groups. The rate of posterior capsule rupture should be very low and did not result a significant effect in the two study groups.